Monomorium (Hymenoptera: Formicidae) of the Arabian Peninsula with description of two new species, M. heggyi sp. n. and M. khalidi sp. n.

http://orcid.org/0000-0001-8137-0705 Sharaf Mostafa R. 1 antsharaf@gmail.com
http://orcid.org/0000-0003-2788-5534 Mohamed Amr A. 2
Boudinot Brendon E. 3
Wetterer James K. 4
http://orcid.org/0000-0003-4709-3083 Hita Garcia Francisco 5
Al Dhafer Hathal M. 1
http://orcid.org/0000-0003-0721-6632 Aldawood Abdulrahman S. 1
1 Department of Plant Protection, College of Food and Agriculture Sciences, King Saud University , Riyadh , Saudi Arabia
2 Department of Entomology, Faculty of Science, Cairo University , Giza , Egypt
3 Department of Entomology and Nematology, University of California , Davis, CA , USA
4 Wilkes Honors College, Florida Atlantic University , FL , USA
5 Biodiversity and Biocomplexity Unit, Okinawa Institute of Science and Technology Graduate University , Onna-son, Okinawa , Japan
Gillespie Joseph
Electronic publication date: 2021 Jan 28
Publication date: 2021
Volume: 9
Electronic Location ID: e10726
Received 2020 Nov 3; Accepted 2020 Dec 16
Copyright: © 2021 Sharaf et al.
Copyright year: 2021
Copyright holder: Sharaf et al.
License: This is an open access article distributed under the terms of the Creative Commons Attribution License, which permits unrestricted use, distribution, reproduction and adaptation in any medium and for any purpose provided that it is properly attributed. For attribution, the original author(s), title, publication source (PeerJ) and either DOI or URL of the article must be cited.
License URL: https://creativecommons.org/licenses/by/4.0/

Keywords: Palearctic region, Afrotropical region

Funding: Ministry of Education, Saudi Arabia IFKSURG-1436-029 The study was funded by the Deputyship for Research & Innovation, “Ministry of Education” in Saudi Arabia through the project no. (IFKSURG-1436-029). The funders had no role in study design, data collection and analysis, decision to publish, or preparation of the manuscript.

==============================
We present a revised and updated synoptic list of 44 Arabian Monomorium species, including two new species of the M. salomonis species-group: M. heggyi sp. n., and M. khalidi sp. n. We propose the following new synonyms: M. abeillei André (= M. wahibiense Collingwood & Agosti syn. n.); M. areniphilum Santschi (= M. fezzanense Collingwood & Agosti syn. n., = M. hemame Collingwood & Agosti syn. n. = M. marmule Collingwood & Agosti syn. n.); M. bicolor Emery (= M. phoenicum Santschi syn. n.); M. harithe Collingwood & Agosti (= M. najrane Collingwood & Agosti syn. n.); M. niloticum Emery (= M. matame Collingwood & Agosti syn. n.); and M. nitidiventre Emery (= M. yemene Collingwood & Agosti syn. n.). An illustrated key and distribution maps are presented for the treated species. Ecological and biological notes are given when available. The majority of Arabian Monomorium species (24) are endemic to the peninsula. All except one of the remaining species are more broadly ranging Afrotropical and Palearctic species, supporting the view of Arabia as a biogeographical crossroads between these two regions. Monomorium floricola (Jerdon), the sole species of Indomalayan origin, is recorded for the first time from the Arabian Peninsula.

Introduction

Monomorium Mayr is one of the largest genera of ants, with 326 valid species and subspecies, as well as three fossil species (Bolton, 2020). Monomorium species predominantly inhabit tropical, subtropical, and warm temperate regions of the world (Bolton, 1987, 1994; Brown, 2000). The majority of Monomorium species are native to the Afrotropical bioregion (Bolton, 1987, 2020; Hita Garcia, Wiesel & Fischer, 2013). In contrast, the Palaearctic and New World faunas are relatively species-poor (Kempf, 1972; Sharaf et al., 2018a; Fernández & Serna, 2019). In total, 56 species have been reported from the Arabian Peninsula (Collingwood, 1985; Collingwood & Agosti, 1996; Aldawood, Sharaf & Collingwood, 2010; Aldawood & Sharaf, 2011; Sharaf & Aldawood, 2013a, 2013b, 2013c; Sharaf, Al Dhafer & Aldawood, 2014; Sharaf, Fisher & Aldawood, 2014; Sharaf et al., 2015, 2017a, 2017b, 2018a, 2018b). Most are generalized feeders or granivores, but some are lestobiotic or parasitic (Ettershank, 1966; Bolton, 1987; Brown, 2000). Notably, Monomorium is one of the few genera that includes several very successful cosmopolitan tramp species, such as M. pharaonis (L.) and M. floricola (Jerdon), M. latinode Mayr, and M. subopacum (Smith) (Bolton, 1987; Wetterer, 2010a, 2010b).

The genus Monomorium has a long and complex taxonomic history (see Heterick, 2006 for a full history of the genus). Emery (1922) offered subgeneric names for several Monomorium species, based on characters of the antennal club and number of antennomeres, some names were later raised to the generic level. Ettershank (1966) made a bold attempt to organize the global fauna of Monomorium and related genera. Bolton (1987) extensively reviewed, refined and keyed the Afrotropical species recognizing 145 species in eight species-groups including descriptions of 46 new species and 43 new synonyms. The Malagasy Monomorium fauna was revised by Heterick (2006) following Bolton’s Afrotropical species-groups as a basis for organizing numerous new species. Heterick (2001, 2003, 2009) comprehensively treated the Australian Monomorium fauna, whereas DuBois (1986) and Mackay & Mackay (2002) studied the Nearctic fauna. The comprehensive phylogenetic work of the subfamily Myrmicinae (Ward et al., 2015) has adopted substantial taxonomic changes including the resurrection of the genera Syllophopsis and Trichomyrmex from synonymy under Monomorium. Sparks, Andersen & Austin (2014) fully revised the Monomorium rothsteini species-complex. Numerous smaller and regional contributions are scattered in the literature including Argentine (Kusnezov, 1949), China (Wu & Wang, 1995), Colombia (Fernández & Serna, 2019), Fiji (Sarnat & Economo, 2012), Italy (Baroni Urbani, 1964, 1968), Japan (Morisita et al., 1992), Morocco (Barech et al., 2017), New Zealand (Brown, 1958), Polynesia (Wilson & Taylor, 1967), Taiwan (Terayama, 2009), the Iberian Peninsula (Collingwood, 1978), Turkmenistan (Dlussky, Soyunov & Zabelin, 1990), and Western Europe (Bernard, 1968).

However, our understanding of the phylogenetic relationships among the currently accepted Monomorium species-groups remains far from satisfactory and the overall taxonomic foundation is in a state of major revision at present (Sparks, Andersen & Austin, 2019). Moreover, our knowledge about the diversity of the genus is still fragmentary for several regions, such as the Mediterranean, the North African-Indian Desert (including what referred as the Saharo-Arabian in some literatures (Holt et al., 2013; Vigna Taglianti et al., 1999; Sharaf et al., 2017a), as well as the Indomalayan and the Neotropical realms, which contain large numbers of described infraspecific taxa in need of further reevaluation (Borowiec, 2014). We note that even the Afrotropical Monomorium fauna, which is the most diverse and best studied so far, is in dire need of an updated taxonomic revision with likely more than 100 undescribed species and many valid species needing reevaluation (Bolton, 1987; Hita Garcia, 2020, personal observation).

The oldest records of Monomorium species from the Arabian Peninsula were of M. carbonarium (Smith) and M. niloticum Emery (Forel, 1907). Collingwood (1985) reported 17 valid Monomorium species from the Kingdom of Saudi Arabia (KSA) and Collingwood & Agosti (1996) listed 49 valid Monomorium species from the entire Arabian Peninsula, including 32 new species. Additional articles have included records from the KSA (Aldawood & Sharaf, 2011; Sharaf & Aldawood, 2013b; Sharaf et al., 2015, 2018a, 2018b), Oman (Sharaf et al., 2018b; Monks et al., 2019), the Socotra Archipelago (Collingwood et al., 2004; Sharaf et al., 2017a), the United Arab Emirates (UAE) (Collingwood et al., 2011), and Yemen (Aldawood, Sharaf & Collingwood, 2010).

Collingwood (1985) and Collingwood & Agosti (1996) recognized a total of 53 Monomorium species on the Arabian Peninsula, of which 17 were described from countries in the region. However, the taxonomic status of several species has remained uncertain due to brief descriptions with insufficient differential diagnoses, apparent ambiguities in the taxonomic keys, and lack of species-group assignment (Sharaf et al., 2020a, 2020b). The present study aims to clarify the current status of the Arabian Monomorium by providing the following: diagnoses of Arabian Monomorium and Monomorium species-groups, a synoptic checklist of Arabian Monomorium, an illustrated identification key to species based on the worker caste, new taxonomic treatments proposing new synonymies and describing two new species, and biogeographical analyses, including distribution maps.

Materials and Methods

The species names follow the online catalogue of ants of the world (Bolton, 2020). We made digital color images of each species using a Leica DFC450 digital camera with a Leica Z16 APO microscope and LAS (v3.8) software. The images are available online on AntWeb (http://www.AntWeb.org) and are accessible through unique specimen identifiers (e.g., CASENT0922288). Distribution maps were made using DIVA-GIS (version 7.5.0.0).

Measurements and indices

All measurements are in millimeters and follow the standard measurements of Bolton (1987) and Sharaf et al. (2018a).

CI = Cephalic Index (HW/HL × 100).

EI = Eye Index (EL/HW × 100).

EL = Eye Length; maximum diameter of eye in lateral view.

EM = Eye-Mandible Distance; distance between anterior margin of eye and mandibular insertion in lateral view.

HL = Head Length; maximum length of head, excluding mandibles in full-face view.

HW = Head Width; maximum width of head directly behind eyes in full-face view.

ML = Mesosoma Length (=Weber Length); length of mesosoma in lateral view); from a point at which pronotum meets cervical shield to posterior base of propodeal lobes or teeth.

PpH = Postpetiole Height; maximum height measured in lateral view.

PpL = Postpetiole Length; maximum length of postpetiolar node measured in dorsal view, from anterior margin to posterior margin.

PpW = Postpetiole Width; maximum width measured in dorsal view.

PtH = Petiole Height; maximum height measured in lateral view.

PtL = Petiole Length; maximum length of petiolar node measured in dorsal view, from anterior margin to posterior margin.

PtW = Petiole Width; maximum width measured in dorsal view.

PW = Pronotal Width; maximum width in dorsal view.

SI = Scape Index (SL/HW × 100).

SL = Scape Length, excluding basal neck.

TL = Total Length, sum of lengths of head, mesosoma, petiole, postpetiole and gaster in profile.

Abbreviations of museums

Abbreviations of natural history collections follow Brandão (2000) except for WML that follows Evenhuis (2020). The material on which this study is based is located and/or was examined at the following institutions:

BMNH The Natural History Museum, London, United Kingdom

CASC California Academy of Sciences, San Francisco, USA

KSMA King Saud University Museum of Arthropods, Riyadh, Kingdom of Saudi Arabia

LACM Los Angeles County Museum of Natural History, Los Angeles, CA, USA

MNHN Muséum National d’Histoire Naturelle, Paris, France

NHMB Naturhistorisches Museum, Basel, Switzerland

OXUM Hope Entomological Collections, University Museum, Parks Road, OXI 3PW Oxford, U.K

WML World Museum Liverpool, Liverpool, U.K

Nomenclatural acts

The electronic version of this article in Portable Document Format (PDF) will represent a published work according to the International Commission on Zoological Nomenclature (ICZN), and hence the new names contained in the electronic version are effectively published under that Code from the electronic edition alone. This published work and the nomenclatural acts it contains have been registered in ZooBank, the online registration system for the ICZN. The ZooBank LSIDs (Life Science Identifiers) can be resolved and the associated information viewed through any standard web browser by appending the LSID to the prefix http://zoobank.org/. The LSID for this publication is: urn:lsid:zoobank.org:pub:A7FFDF5C-6CD5-41CA-B106-BFF6BDFCB258; for Monomorium heggyi sp. n. is urn:lsid:zoobank.org:act:B57EC2EA-1781-4C19-ADFE-757C834E2774; and for Monomorium khalidi sp. n. is urn:lsid:zoobank.org:act:3B5BB529-D842-4146-B8F7-ACCAC9CD5BA7. The online version of this work is archived and available from the following digital repositories: PeerJ, PubMed Central and CLOCKSS.

Results

Diagnosis of Arabian Monomorium

Members of the genus Monomorium can be recognized by combination of the following characters in the worker caste (Bolton, 1987, 1994; DuBois, 1986; Fisher & Bolton, 2016; Sparks, Andersen & Austin, 2019): monomorphic to polymorphic; antennae 10–12 segmented (most frequently 12), conspicuous 3-segmented club; mandibles with 3–4 teeth; palp formula 2,2, or 1,2; median clypeal seta conspicuous; median portion of clypeus raised, the raised portion usually longitudinally bicarinate; lateral portions of the clypeus not elevated as shield-like ridges anterior to the antennal toruli; frontal carinae distinct, but absent posterior to the medial arches of the antennal toruli; metanotal groove present, commonly impressed; propodeal dorsum usually unarmed and rounding into the declivity; propodeal spiracle usually circular, located at about the midlength of the sclerite; abdominal segment IV (metasomal III, gastral I) with posttergite overlapping poststernite.

Diagnosis of Arabian Monomorium species-groups

The M. monomorium species-group can be readily recognized by the following combination of characters in the worker caste (Sharaf et al., 2018a): monomorphic, with size variation; head longer than broad; mandibles smooth and masticatory margin armed with four teeth; antennae with 10–12 segments, terminating in a three-segmented club; median clypeal portion raised anteriorly and longitudinally bicarinate; eyes present with variable size, located in front of the midlength of the sides in full-face view, and with four or more ommatidia in the longest row; head smooth and shining; metanotal groove well-defined, with distinct cross-ribs; propodeal spiracle circular to subcircular; propodeal dorsum meeting declivity in a rounded angle; promesonotum and propodeal dorsum smooth; body pilosity variable but usually distinct; petiole, postpetiole and gastral tergites usually smooth.

The M. salomonis species-group can be diagnosed by the following character states in the worker caste (Bolton, 1987): monomorphic, with minor size variation; palp formula 2,2 or 1,2 in some minute species; mandibles usually sculptured; masticatory margins of mandibles armed with four teeth which decrease in size from apex to base; median clypeal portion raised, projecting anteriorly; cephalic dorsum usually sculptured, ranging from dense blanketing reticulate-punctuation to feeble superficial reticular patterning; eyes prominent of medium to large size, usually with six or more ommatidia in the longest row; eyes circular to oval in shape; head longer than broad; scapes usually relatively long (SI > 80); metanotal groove moderately impressed to absent; metanotal cross-ribs inconspicuous to absent; propodeal spiracle circular to subcircular; propodeum rounded to angular between dorsum and declivity; propodeal dorsum usually sculptured but never transversely striate; petiolar spiracle situated at the node or immediately in front of the anterior face of the node; body pilosity variable in distribution and density, but usually reduced on the head and mesosoma; mesosoma, petiole and postpetiole usually sculptured; first gastral tergite usually shagreenate or finely sculptured.

Synoptic checklist of Arabian Monomorium

Monomorium monomorium species-group

1. Monomorium aeyade Collingwood & Agosti, 1996

2. Monomorium brunneolucidulum Collingwood & Agosti, 1996 (nomen dubium).

3. Monomorium carbonarium (Smith, 1858)

  = Monomorium minutum Mayr, 1862

4. Monomoriumclavicorne André, 1881

  = Monomorium clavicorne punicum Santschi, 1915a

5. Monomorium exiguum Forel, 1894

  = Monomorium exiguum bulawayensis Forel, 1913

  = Monomorium faurei Santschi, 1915b

  = Monomorium exiguum flavescens Forel, 1916

  = Monomorium baushare Collingwood & Agosti, 1996

  = Monomorium qarahe Collingwood & Agosti, 1996

6. Monomorium floricola (Jerdon, 1851)

  = Monomorium poecilum Roger, 1863

  = Monomorium cinnabari Roger, 1863

  = Monomorium specularis Mayr, 1866

  = Monomorium impressum Smith, 1876

  = Monomorium floreanum Stitz, 1932

  = Monomorium angusticlava Donisthorpe, 1947

7. Monomorium holothir Bolton, 1987

8. Monomorium mohammedi Sharaf & Hita Garcia, 2018

9. Monomorium rimae Collingwood & Agosti, 1996

10. Monomorium sarawatense Sharaf & Aldawood, 2013b

Monomorium salomonis species-group

11. Monomorium abeillei André, 1881

  = Monomorium wahibiense Collingwood & Agosti, 1996 syn. n.

12. Monomorium acutinode Collingwood & Agosti, 1996

13. Monomorium areniphilum Santschi, 1911

  = Monomorium fezzanense Collingwood & Agosti, 1996 syn. n.

  = Monomorium hemame Collingwood & Agosti, 1996 syn. n.

  = Monomorium marmule Collingwood & Agosti, 1996 syn. n.

  = Monomorium salomonis lepineyi Santschi, 1934

  = Monomorium salomonis pullula Santschi, 1919

14. Monomorium asiriense Collingwood & Agosti, 1996

15. Monomorium barbatulum Mayr, 1877 (New record to KSA)

16. Monomorium bicolor Emery, 1877

  = Monomorium bicolor aequatoriale Santschi, 1926

  = Monomorium bicolor coerulescens Santschi, 1912

  = Monomorium bicolor rufibasis Santschi, 1914

  = Monomorium bicolor tropicale Santschi, 1926

  = Monomorium bicolor uelense Santschi, 1926

  = Monomorium phoenicum Santschi, 1927 syn. n.

17. Monomorium buettikeri Collingwood & Agosti, 1996

18. Monomorium buxtoni Crawley, 1920a

19. Monomorium carbo Forel, 1910

20. Monomorium dammame Collingwood & Agosti, 1996

21. Monomorium dirie Collingwood & Agosti, 1996

22. Monomorium elghazalyi Sharaf & Aldawood, 2017

23. Monomorium fayfaense Collingwood & Agosti, 1996

24. Monomorium gallagheri Collingwood & Agosti, 1996

25. Monomorium hanaqe Collingwood & Agosti, 1996

26. Monomorium harithe Collingwood & Agosti, 1996

  = Monomorium najrane Collingwood & Agosti, 1996 syn. n.

27. Monomorium heggyi Sharaf, sp. n.

28. Monomorium jizane Collingwood & Agosti, 1996

29. Monomorium khalidi Sharaf, sp. n.

30. Monomorium knappi Collingwood & Agosti, 1996

31. Monomorium luteum Emery, 1881

32. Monomorium mahyoubi Collingwood & Agosti, 1996

33. Monomorium moathi Sharaf & Collingwood, 2010

34. Monomorium niloticum Emery, 1881

  = Monomorium matame Collingwood & Agosti, 1996 syn. n.

35. Monomorium nimihil Collingwood, 2004

36. Monomorium nitidiventre Emery, 1893

  = Monomorium yemene Collingwood & Agosti, 1996 syn. n.

37. Monomorium pharaonis (Linnaeus, 1758)

  = Monomorium antiguensis (Fabricius, 1793)

  = Monomorium domestica (Shuckard, 1838)

  = Monomorium contigua (Smith, 1858)

  = Monomorium fragilis (Smith, 1858)

  = Monomorium minuta (Jerdon, 1851)

  = Monomorium vastator (Smith, 1857)

38. Monomorium riyadhe Collingwood & Agosti, 1996

39. Monomorium salomonis (Linnaeus, 1758)

  = Monomorium debilis (Walker, 1871)

  = Monomorium salomonis obscuratum Stitz, 1917

  = Monomorium thorense Mayr, 1862

40. Monomorium subdenticorne Collingwood & Agosti, 1996

41. Monomorium subopacum (Smith, 1858), full synonymy in Heterick (2006)

42. Monomorium suleyile Collingwood & Agosti, 1996

43. Monomorium tumaire Collingwood & Agosti, 1996

44. Monomorium venustum (Smith, 1858)

Key to Arabian Monomorium based on the worker caste

Note: M. brunneolucidulum excluded due to lack of diagnostic characters.

1. Head smooth and glossy (Fig. 1A)2 (M. monomorium species-group)

– Head sculptured, sculpture ranging from dense reticulate-punctate, longitudinal striations, to faint reticular patterning (Fig. 1B)10 (M. salomonis species-group)

2. Anterior median portion of clypeus strongly concave with two laterally projecting teeth (Fig. 1C)3

– Anterior median portion of clypeus straight or feebly concave without raised ridges (Fig. 1D)4

3. Uniform dark brown to black; metanotal groove broadly and deeply impressed (Fig. 1E)M. carbonarium

– Bicolored with yellow mesosoma contrasting with the black gaster; metanotal groove shallowly impressedM. rimae

4. Antenna 12-segmented5

– Antenna 11-segmented7

5. Body pilosity clubbed; mesosoma, petiole and postpetiole distinctly sculptured (Fig. 1F)M. sarawatense

– Body pilosity simple; mesosoma, petiole and postpetiole smooth and shining (Fig. 2A)6

6. Bicolored, with head and gaster dark brown to black contrasting the yellow or yellow-brown mesosoma and petiole; eyes small (EL 0.21–0.24 x HW) with six ommatidia in longest row, oval in profile (Fig. 2A)M. floricola

– Uniform yellow; eyes relatively large (EL 0.30–0.32 x HW) with 8-9 ommatidia in longest row, in profile with convex dorsal margin and straight ventral margin (Fig. 2B)M. holothir

7. Mesosoma without standing hairs (Fig. 2C)8

– Mesosoma with standing hairs (Fig. 2D)9

8. Eyes larger, with a ring of seven to eight ommatidia encircling a single row of 2 ommatidia, and in profile closer to mandibular insertions (EM 0.05); meso- and metapleuron smooth; petiole and postpetiole smooth and each with one pair of standing hairs (Fig. 2C)M. aeyade

– Eyes smaller, with only 5–6 ommatidia total, and in profile further away from mandibular insertions (EM 0.09–0.11); meso-and metapleuron finely shagreened; petiole and postpetiole superficially shagreened and without standing hairs (Fig. 2E)M. mohammedi

9. Mesosoma with two pairs of standing hairs, one on pronotal corners and one on propodeum (Fig. 2F)M. clavicorne

– Mesosoma with several pairs of standing hairs, about 10 pairs (Fig. 2D)M. exiguum

10. Underside of head with crowded J-shaped hairs forming a distinct psammophore (Fig. 3A)M. barbatulum

– Underside of head without long J-shaped hairs, psammophore absent11

11. Mesosoma without standing hairs12

– Mesosoma with standing hairs25

12. Uniform yellow, brown, or yellow-brown, gaster not darker than mesosoma13

– Bicolored, with gaster distinctly darker than mesosoma18

13. Metanotal groove shallowly impressed or indistinct (Fig. 3B)14

– Metanotal groove deeply impressed (Fig. 3C)15

14. Small yellow species (TL 1.7, HL 0.45–0.48, HW 0.34–0.36, SI 92–97); cephalic surface with vestigial or superficial reticular patterning, almost entirely effaced; petiole and postpetiole without standing hairs; (Fig. 3B); first gastral tergite completely glabrous, that is, without standing hairs (Fig. 3B)M. heggyi sp. n.

– Large brown species (TL 2.20–2.75, HL 0.57–0.65, HW 0.51, SI 93–103); cephalic surface with fine and dense reticulate-rugulose sculpture; petiole and postpetiole each with two pairs of standing hairs (Fig. 3D); first gastral tergite always with several pairs of standing hairs (Fig. 3D)M. harithe

15. Eyes located nearly at the midlength of head as seen in full-face view (Fig. 3E); eyes smaller, with 5 ommatidia in longest row; posterior margin of head distinctly concave in full-face view (Fig. 3E)M. elghazalyi

– Eyes located nearly behind or at head midlength as seen in full-face view (Fig. 3F); eyes distinctly larger, with 10–14 ommatidia in longest row; posterior margin of head concave or linear in full-face view16

16. Underside of head without long, standing hairs (Fig. 4A)M. dirie

– Underside of head with numerous pairs of long, standing hairs (Fig. 4B)17

17. Yellow; scapes just reaching posterior margin of head when laid back; body pilosity abundant over entire bodyM. nimihil

– Brown; scapes surpassing posterior margin of head by about half the length of the pedicel when laid back; body pilosity distinctly reduced over entire surface, mesosoma with a single pair of hairs on propodeum, while the petiole, postpetiole, and gaster are bareM. salomonis

18. First gastral tergite without standing hairsM. moathi

– First gastral tergite with hairs either scattered on tergite surface or apically on the posterior margin19

19. Scapes when laid back from their insertions reach or surpass posterior margin of head in full-face view20

– Scapes when laid back from their insertions fail to reach posterior margin of head in full-face view21

20. Propodeal dorsum in profile meeting declivity forming two blunt, slightly projecting angled bosses (Fig. 4C)M. subdenticorne

– Propodeal dorsum in profile meeting declivity in a continuous curve (Fig. 4D)M. bicolor

21. Petiole in the form of a high triangle in profile with anterior face appears as a continuous line sloping anteriorly (Fig. 4E)M. acutinode

– Petiole broadly rounded with anterior face sloping downward and then anteriorly to the peduncle (Fig. 4F)22

22. Head faintly superficial sculptured, slightly shiningM. venustum

– Head densely and finely reticulate to reticulate-shagreenate23

23. Small (TL 2.3–2.4, HL 0.60–0.63, HW 0.43–0.45, PW 0.30–0.31, ML 0.66–0.70); mesosoma and gaster approximately of the same color, the two not strongly contrastingM. carbo

– Larger (TL 3.1–3.4, HL 0.70–1.04, HW 0.54–0.88, PW 0.36–0.53, ML 0.88–1.24); mesosoma and gaster conspicuously differently colored, gaster usually darker24

24. Eyes with 12–14 ommatidia in longest row; metanotal groove deeply impressed (Fig. 5A); postpetiole with two to three pairs of backward directed hairsM. areniphilum

– Eyes with 9–11 ommatidia in longest row; metanotal groove feebly impressed (Fig. 5B); postpetiole with a single pair of backward directed hairsM. subopacum

25. Large (TL ≥ 3.8, HW > 0.75)26

– Smaller (TL 2.2–3.2, ≤ HW 0.67)28

26. Entirely yellowish; mesosoma rather flat with a shallow oblique metanotal groove (Fig. 5C)M. luteum

– Gaster dark contrasting with red mesosoma; metanotal groove steeply angled (Fig. 5D)27

27. Head smooth with superficial sculpture; the first of the three segments forming the club being shorter than the second (Fig. 5E); head in full-face view with feebly but distinctly convex sides; head in full-face view with eyes fail or just break head sides (Fig. 5E)M. niloticum

– Head completely finely striate (Fig. 5F); the first of the three segments forming the club nearly subequal to the second (Fig. 5F); head in full-face view with straight sides; head in full-face view with eyes break sides (Fig. 5F)M. riyadhe

28. Mesosoma red, contrasting with dark gaster29

– Mesosoma pale brown to black, concolorous with gaster34

29. Mesosoma hairs restricted to one pair on pronotum30

– Mesosoma with several pairs of hairs scattered over whole dorsum31

30. Head and mesosoma dull red, gaster brown; scapes reaching posterior head margin when laid back (Fig. 6A); cephalic surface with vestigial sculptures; posterior margin of head with a single pair of hairs (Fig. 6A); propodeum in profile with dorsum making an obtuse angle with declivity (Fig. 6B)M. hanaqe

– Head and mesosoma bright orange red, gaster black; scapes not reaching posterior head margin when laid (Fig. 6C); cephalic surface densely punctate; posterior margin of head without hairs except for appressed pubescence (Fig. 6C); propodeum in profile with dorsum making a continuous curve with declivity (Fig. 6D)M. jizane

31. Propodeal dorsum with a single pair of standing hairs32

– Propodeal dorsum with at least three pairs of standing hairs33

32. Scapes when laid back from their insertions reach posterior margin of head in full-face view (Fig. 6E); cephalic surface densely punctate (Fig. 6E); gaster smooth and shining (Fig. 6F)M. fayfaense

– Scapes when laid back from their insertions fail to reach posterior margin of head in full-face view (Fig. 7A); cephalic surface with vestigial sculpture (Fig. 7A); gaster finely densely shagreened and dull (Fig. 7B)M. knappi

33. Whole body with abundant fine hairs (Fig. 7C); with head in full-face view outer margins of eyes break head sides (Fig. 7D)M. nitidiventre

– Body pilosity limited and stiff (Fig. 7E); promesonotum with five to six pair of standing hairs, promesonotum and propodeum each with three pairs; petiole and postpetiole each with two-three pairs; with head in full-face view outer margins of eyes fail to break head sides (Fig. 7F)M. khalidi sp. n.

34. Whole body including gaster densely sculptured and dull35

– Gaster at least more or less shining with superficial sculpture37

35. Head, mesosoma, and waist segments very light brown or yellow; hairs on mesosoma scattered (Fig. 8A)M. pharaonis

– Head, mesosoma, and waist segments conspicuously of darker brown or black; mesosoma hairs mainly or entirely on pronotum (Fig. 8B)36

36. Body color brown; larger species (TL 3.1–3.3, HW 0.70); scapes long, surpassing posterior head margin by about the length of the pedicel when laid back (Fig. 8C); head in full-face view with convex sides (Fig. 8C)M. suleyile

– Body color uniformly black; smaller species (TL 2.7–2.9, HW 0.56); scapes shorter, just reaching posterior head margin when laid back of head (Fig. 8D); head in full-face view with parallel sides (Fig. 8D)M. mahyoubi

37. Whole body glossy, nodes and gaster brilliantM. dammame

– At least mesosoma with close punctate sculpture38

38. Underside of head with numerous hairs (12–16), the longest exceeding the maximum eye length (Fig. 8E)M. tumaire

– Underside of head with fewer hairs, none as long as maximum eye length (Fig. 8F)39

39. Head finely and densely punctate, general appearance dull40

– Head feebly superficially sculptured, relatively but distinctly shining41

40. Smaller species (TL 2.4–2.7); clypeal border feebly concave to straight with a very small median notch or none (Fig. 9A); mesonotum straight in profile (Fig. 9B)M. gallagheri

– Larger species (TL > 3.0); clypeus with a distinct anteromedian notch (Fig. 9C); mesonotum distinctly convex in profile (Fig. 9D)M. buxtoni

41. Scapes distinctly short, surpassing compound eye posterior margins by only about the length of the first funicular segmentM. buettikeri

– Scapes longer, reaching posterior head margin42

42. Color dark brown to black; smaller species (TL 2.40–2.75, C1 78–91); metanotal groove shallowly impressedM. abeillei

– Color light to median brown; larger species (TL 3.2, CI [69–71]); metanotal groove distinctly impressedM. asiriense

Figure 1 (A) Head of M. exiguum in full-face view, CASENT0217367 (Erin Prado); (B) head of M. khalidi, sp. n. in full-face view, CASENT0922288 (Michele Esposito); (C) head of M. carbonarium in full-face view, CASENT0902279 (Ryan Perry); (D) head of M. floricola in full-face view, CASENT0922876 (Michele Esposito); (E) body of M. carbonarium in profile, CASENT0902279 (Ryan Perry); (F) body of M. sarawatense in profile, CASENT0280971 (Estella Ortega), www.AntWeb.org, licensed under CC BY 3.0 Unported.

Figure 2 (A) Body of M. floricola in profile, CASENT0922876 (Michele Esposito); (B) body of M. holothir in profile, CASENT0902243 (Will Ericson); (C) body of M. aeyade in profile, CASENT0922329 (Michele Esposito); (D) body of M. exiguum. In profile, CASENT0217367 (Erin Prado); (E) body of M. mohammedi in profile, CASENT0922351 (Michele Esposito); www.AntWeb.org, (F) body of M. clavicorne in profile, (Francisco Hita Garcia). Key illustrations.

Figure 3 (A) Head of M. barbatulum in profile, CASENT0922263 (Michele Esposito); (B) mesosoma of M. rabirium in profile, CASENT0746641 (Zach Lieberman); (C) mesosoma of M. elghazalyi in profile, CASENT0746626 (Michele Esposito); (D) petiole and postpetiole of M. harithe in profile, CASENT0913802 (Will Ericson); (E) head of M. elghazalyi in full-face view, CASENT0746626 (Michele Esposito); (F) head of M. dirie in full-face view, CASENT0913571 (Alexandra Westrich), www.AntWeb.org, licensed under CC BY 3.0 Unported. Key illustrations.

Figure 4 (A) Head of M. dirie in profile, CASENT0913571 (Alexandra Westrich); (B) head of M. salomonis in profile, CASENT0913835 (Will Ericson); (C) mesosoma of M. subdenticorne in profile, CASENT0914318 (Zach Lieberman); (D) mesosoma of M. bicolor in profile, CASENT0073615 (Michele Esposito); (E) petiole of M. acutinode in profile, CASENT0913547 (Will Ericson); (F) petiole of M. carbo in profile, CASENT0249908 (Shannon Hartman), www.AntWeb.org, licensed under CC BY 3.0 Unported. Key illustrations.

Figure 5 (A) Mesosoma of M. areniphilum in profile, CASENT0048600 (Michele Esposito); (B) mesosoma of M. subopacum in profile, CASENT0064820 (April Nobile); (C) mesosoma of M. luteum in profile, CASENT0904599 (Will Ericson); (D) mesosoma of. M. niloticum in profile, CASENT0260164 (Estella Ortega); (E) head of M. niloticum in full-face view, CASENT0919811 (Michele Esposito); (F) head of M. riyadhe in full-face view, CASENT0922342 (Michele Esposito), www.AntWeb.org, licensed under CC BY 3.0 Unported. Key illustrations.

Figure 6 (A) Head of M. hanaqe in full-face view, CASENT0249834 (Ryan Perry); (B) mesosoma of M. hanaqe in profile, CASENT0249834 (Ryan Perry); (C) head of M. jizane in full-face view, CASENT0913806 (Will Ericson); (D) propodeum of M. jizane in profile, CASENT0913806 (Will Ericson); (E) head of M. fayfaense in full-face view, CASENT0249833 (Ryan Perry); (F) gaster of M. fayfaense in profile, CASENT0249833 (Ryan Perry), www.AntWeb.org, licensed under CC BY 3.0 Unported. Key illustrations.

Figure 7 (A) Head of M. knappi in full-face view, CASENT0913812 (Will Ericson); (B) body of M. knappi in profile, CASENT0913812 (Will Ericson); (C) body of M. nitidiventre in profile, CASENT0904602 (Will Ericson); (D) head of M. nitidiventre in. full-face view, CASENT0904602 (Will Ericson); (E) body of M. khalidi sp. n. in profile, CASENT0922288 (Michele Esposito); (F) head of M. khalidi sp. n. in profile, CASENT0922288 (Michele Esposito), www.AntWeb.org, licensed under CC BY 3.0 Unported. Key illustrations.

Figure 8 (A) Mesosoma of M. pharaonis in profile, CASENT0246072 (Andrea Walker); (B: mesosoma of M. buxtoni in profile, CASENT0902220 (Will Ericson); (C) head of M. suleyile in full-face view, CASENT0913854 (Zach Lieberman); (D) head of. M. mahyoubi in full-face view, CASENT0913823 (Alexandra Westrich); (E) head of M. tumaire in profile, CASENT0249858 (Ryan Perry); (F) head of M. buettikeri in profile, CASENT0913565 (Zach Lieberman), www.AntWeb.org, licensed under CC BY 3.0 Unported. Key illustrations.

Figure 9 (A) Head of M. gallagheri in full-face view, CASENT0913582 (Zach Lieberman); (B) head of M. buxtoni in full-face view, CASENT0902220 (Zach Lieberman); (C) body of M. abeillei in profile, CASENT0915411 (Will Ericson); (D) body of. M. asiriense in profile, CASENT0913560 (Zach Lieberman), www.AntWeb.org, licensed under CC BY 3.0 Unported. Key illustrations.

New taxonomic treatments

New site records include, when available, geo-coordinates (°N, °E), elevation (m), collection date, collector, and number worker (w) and queen (q) specimens.

Monomorium abeilleiAndré, 1881

(Figs. 10A–10C)

Figure 10 M. abeillei, (A) body in profile; (B) body in dorsal view; (C) head in full-face view, CASENT0915411 (Will Ericson), www.AntWeb.org, licensed under CC BY 3.0 Unported.

Monomorium abeillei André, 1881: 531 (footnote) (w.) Israel. Palearctic. (MNHN, CASENT0915411). [Image of lectotype worker examined]. [Also described as new by André, 1881: 67.].

Combination in Monomorium (Xeromyrmex): Emery, 1922: 177; subspecies of Monomorium salomonis: Forel, 1910a: 23; Hamann & Klemm, 1967: 413; revived status as species: Collingwood, 1985: 269; Collingwood & Agosti, 1996: 340.

Monomorium wahibiense Collingwood & Agosti, 1996: 357 (w.) Oman. Palearctic. [NHMB], CASENT0913864. Syntype worker, Oman, Wahiba dunes (21.vii.1985, M. D. Gallagher) [examined] syn. n.

Material examined. KSA: Asir Province: Raydah (18.198, 42.410, 2,443 m, 22.ii.2014, M.R. Sharaf, 3w); Raydah (18.204, 42.412, 2,820 m, 21.ii.2014, M.R. Sharaf, 6w); Riyadh Province: Huraymila, Buaythiran (25.149, 45.950, 07.ii.2011, M.R. Sharaf, 2w); Rawdhat Khureim (25.383, 47.277, 618 m, 02.vi.2013, S. Salman, 6w); Rawdhat Khureim (25.425, 47.235, 579 m, 09.i.2015, S. Salman, 12w); Dirab, KSU research station (24.419, 46.654, 568 m, 05.xii.2013, S. Salman, 1w); Wadi Hanifa (24.670, 46.654, 657 m, 14.ii.2014, S. Salman, 3w); Mezahmyia (24.472, 46.239, 633 m, 25.i.2014, S. Salman, 1w); Mezahmyia (24.466, 46.251, 648 m, 29.xi.2014, S. Salman, 1w); Al Hayer (24.546, 46.742, 647 m, S. Salman, 2w); Al Hayer (24.557, 46.744, 589 m, 11.iv.2014, S. Salman, 6w); Runnah (25.571, 46.973, 615 m, 12.iv.2014, S. Salman, 1w); Dawademi (24.478, 44.364, 1,027 m, 18.iv.2014, S. Salman, 1w); Dawademi (24.583, 44.323, 966 m, 16.i.2015, S. Salman, 3w); Dawademi (24.538, 44.355, 999 m, 16.i.2015, S. Salman, 1w); Afif (23.766, 42.840, 1,015 m, 19.iv.2014, S. Salman, 6w); Afif (24.302, 43.688, 951 m, 19.iv.2014, S. Salman, 1w); Afif (23.900, 42.081, 1,052 m, 17.i.2015, S. Salman, 15w); Afif (23.957, 42.976, 1,059 m, 17.i.2015, S. Salman, 1w); Irgah (24.6710, 46.593, 625 m, 19.i.2015, S. Salman, 2w); Thadiq (25.294, 45.871, 735 m, 26.iv.2014, S. Salman, 1w); Quwayia (24.047, 45.244, 854 m, S. Salman, 7w); Shaqra (25.326, 45.233, 710 m, 30.v.2014, S. Salman, 9w); Shagra (25.230, 45.319, 703 m, 24.i.2015, S. Salman, 1w); Shaqra (25.270, 45.291, 712 m, 23.i.2015, 2w); Durma (24.607, 46.130, 646 m, 30.i.2015, S. Salman, 1w); Majma’a (25.880, 45.365, 730 m, 07.ii.2015, S. Salman, 5w); Kharrarah (24.392, 46.244, 726 m, 08.iv.2015, S. Salman, 15w); Al Ghat (26.066, 44.919, 653 m, 31.x.2015, S. Salman, 1w); Hawtet Sudeir (25.592, 45.612, 732 m, 31.i.2015, S. Salman, 1w); KSU campus (24.737, 46.618, 662 m, 29.ii.2012, K. Mahmoud, 1w); Dirab (24.419, 46.654, 804 m, 18.ix.2014, S. Salman, 9w); Hareeq (23.614, 46.054, 689 m, 22.ii.2015, S. Salman, 1w); Quwayia (24.058, 45.245, 846 m, 29.xi.2014, S. Salman, 1w); Salboukh (25.078, 46.347, 716 m, 26.xii.2014, S. Salman, 8w); Jazan Province: Sajid Island, Al-Sajid (16.860, 41.932, 05.iii.2017, U. Abuelgheit, 1w).

Remarks. Monomorium wahibiense is represented by a single worker deposited in WML and accompanied by a red card and handwritten label by C. Collingwood indicating that this specimen represents the syntype. The label’s data are consistent with the data for the type in the original description in terms of collecting locality (Oman, Wahiba sand) and collector (M.D. Gallagher), but not the collection date, which we consider a typographical error.

Comparing the mentioned type material with the image of the type material of M. abeillei André, we found the two species share the same morphological characters, which can be summarized as follow: scapes relatively short, when laid back from their insertions just reaching posterior head margin; eyes of moderate size with about 10–11 ommatidia in longest row; cephalic surface between frontal lobes faintly striated whereas in some individuals the striations are absent; promesonotum and mesonotum forming continuous flat line in profile; mesosoma with single pair of standing hairs on pronotal humeral angles; metanotal groove feebly impressed; petiole with single pair of backward directed hairs; postpetiole with two pairs. However, the eyes are slightly smaller in M. abeillei, and the central cephalic sculpture is feebly microreticulate-striolate than in the relatively large-eyed M. wahibiense and the superficially sculptured cephalic surface but we consider these two traits as variable characters. Herein, we propose treating M. wahibiense as a junior synonym of M. abeillei André.

Geographic Distribution. Monomorium abeillei is originally described from Israel and recorded from several countries in the Middle East, including the Arabian Peninsula (KSA, Kuwait, Oman and Yemen) (Collingwood, 1985, Collingwood & Agosti, 1996), Iran (Paknia et al., 2008), Israel (Vonshak & Ionescu-Hirsch, 2009) and North Africa (Borowiec, 2014).

Monomorium areniphilum Santschi, 1911

(Figs. 11A–11C)

Figure 11 M. areniphilum, (A) body in profile; (B) body in dorsal view; (C) head in full-face view, CASENT0048600 (Michele Esposito), www.AntWeb.org, licensed under CC BY 3.0 Unported.

Monomorium salomonis var. areniphila Santschi, 1911: 84 (w.) Tunisia. Palearctic. [NHMB], CASENT0249829, [Syntype worker, examined].

Emery, 1915: 378 (q.); combination in Monomorium (Xeromyrmex): Emery, 1922: 177; subspecies of Monomorium salomonis: Santschi, 1936: 50; raised to species: Collingwood, 1985: 269; senior synonym of Monomorium lepineyi, Monomorium pullula: Bolton, 1987: 336.

Senior synonym of Monomorium fezzanense Collingwood & Agosti, 1996: 346 (w.) Saudi Arabia. Afrotropic. [NHMB], Syntype worker, Saudi Arabia, 31 km NW Tabuk (24.iv.1979, CASENT0913557) [examined] syn. n.

Senior synonym of Monomorium hemame Collingwood & Agosti, 1996: 348 (w.) Kuwait. Palearctic. [WML], holotype worker, Kuwait, Umm Al-Hemam (9.III.1988, W. Biittiker, CASENT0922316) [examined]; paratype worker, Saudi Arabia, Uyaynah (01.IV 1976, W. Biittiker, CASENT0913800) [examined] syn. n.

Senior synonym of Monomorium marmule Collingwood & Agosti, 1996: 349, fig. 21 (w.) OMAN. Palearctic. Paratype worker, Oman, Minririb (14.i.1986, M.D. Gallagher, CASENT0913824, NHMB) [examined] syn. n.

Remarks. A thorough examination of the type material of M. fezzanense, M. hemame, M. marmule, and M. areniphilum yielded no evidence for heterospecificity; they are indistinguishable. All four taxa share the following characters: median portion of anterior clypeal margin shallowly concave; eyes large with 12–15 ommatidia in longest row; promesonotum and anterior portion of mesonotum in profile feebly convex; posterior portion of mesonotum sloping steeply to broadly and deeply impressed metanotal groove; mesosoma without hairs; petiole with single pair of backward directed hairs; postpetiole with two pairs of hairs.

Herein M. fezzanense, M. hemame, and M. marmule are treated as junior synonyms of M. areniphilum. It is worth mentioning that in the original description of M. marmule, Collingwood & Agosti (1996) gave a brief differential diagnosis with M. areniphilum based on variable characters such as the presence of mesosomal pubescence, the petiole and postpetiole color and pilosity.

Geographic Distribution. A species originally described from Tunisia and recorded from most countries of the Arabian Peninsula including KSA, Kuwait, Oman, and Yemen (Collingwood, 1985; Collingwood & Agosti, 1996), and the UAE (Collingwood et al., 2011). It is also reported from North Africa and the Afrotropical Region (Bolton, 1987).

Monomorium barbatulum Mayr, 1877

(Figs. 12A–12C)

Figure 12 M. barbatulum, (A) body in profile; (B) body in dorsal view; (C) head in full-face view, CASENT0922263 (Michele Esposito), www.AntWeb.org, licensed under CC BY 3.0 Unported.

Monomorium barbatulum Mayr, 1877: 17 (w.) Kazakhstan. Palearctic.

Material examined. KSA: Riyadh Province: Zulfi (26.367, 44.986, 670 m, 18.i.2014, Al Dhafer et al., 1w: CASENT0922263, KSMA).

Geographic Distribution. This species was originally described from Kazakhstan and recorded from Oman (Collingwood, 1985; Collingwood & Agosti, 1996), the UAE (Collingwood et al., 2011), Turkey (Kiran & Karaman, 2020), Israel (Vonshak & Ionescu-Hirsch, 2009). The present material represents a new record to the KSA.

Remarks. Monomorium barbatulum looks similar to some members of the genus Trichomyrmex in terms of the following characters: polymorphic with 12-segmented antennae that lacking the well-defined terminal club; masticatory margin of mandibles armed with 3–4 teeth; propodeum unarmed. However, M. barbatulum lacks a critical diagnostic character for the genus Trichomyrmex which is the absence of the transverse striations on propodeal dorsum. With the availability of more material for a comprehensive taxonomic investigation, together with additional molecular evidence(s), we will be able to resolve properly the taxonomic status of the species.

Monomorium bicolorEmery, 1877

(Figs. 13A–13C)

Figure 13 M. bicolor, (A) body in profile; (B) body in dorsal view; (C) head in full-face view, CASENT0904601 (Will Ericson), www.AntWeb.org, licensed under CC BY 3.0 Unported.

Monomorium bicolor Emery, 1877: 368 (w.) Eritrea. Afrotropic.

Senior synonym of Monomorium phoenicum Santschi, 1927e: 242 (w.q.) Lebanon. Palearctic. Syntype worker, Lebanon, Beyroth (05.viii.1933, Santschi, CASENT0249831, NHMB) [examined] syn. n.

Material examined. KSA: Asir Province: Almajardah, Wadi Eltalalei (19.003, 41.732, 223 m, 10.xi.2012, M.R. Sharaf, 1w); Wadi Shahadan (17.472, 42.856, 452 m, 13.xi.2012, M.R. Sharaf, 13w, (1w, CASENT0906396); Allaith, Adam, Wadi Elarj (20.453, 40.816, 450 m, 09.xi.2012, M.R. Sharaf, 45w, 1w: CASENT0906395, 1q: CASENT0906394); Wadi Aljora, near Abadan (17.293, 43.070, 465 m, 12.xi.2012, M.R. Sharaf, 2w); Almajardah, Wadi Bagara (18.793, 42.019, 436 m, 10.xi.2012, M.R. Sharaf, 2w); Jazan Province: Abu Arish (17.013, 42.802, 90 m, 10.iv.2012, M.R. Sharaf, 11w); Sabia (17.107, 42.650, 43 m, 09.iv.2012, M.R. Sharaf, 23w); Jazan (16.97627, 42.61743, 38 m, 12.iv.2012, M.R. Sharaf, 12w); Al Bahah Province: Dhi Ayn Archeological village (19.930, 41.443, 741 m, 18.v.2011, M.R. Sharaf, 3w); Wadi Gonouna (19.429, 41.605, 353 m, 12.v.2011, M.R. Sharaf, 10w: KSMA).

Remarks. The type material of M. bicolor and M. phoenicum are clearly conspecific. They share the same diagnostic characters as follow: scapes relatively long, when laid back from their insertions surpassing posterior head margin by about length of pedicel; head in full-face view with eyes just breaking sides; cephalic surface dull, finely and densely punctate; median anterior clypeal margin distinctly concave; area between frontal carinae finely longitudinally striated; mesosoma without standing hairs; metanotal groove acutely impressed; propodeal dorsum in profile meeting declivity in continuous curve; petiole and postpetiole each with single pair of back directed hairs; first gastral tergite with hairs scattered over tergite surface; biocolored species, with head, mesosoma, petiole and postpetiole yellow-red or yellow-brown, gaster dark brown to black. Herein, we propose M. phoenicum as a junior synonym of M. bicolor.

Geographic Distribution. Monomorium bicolor was originally described from Eritrea and is a widespread species commonly encountered in open, sandy areas through the Afrotropical Region (Bolton, 1987). In the Arabian Peninsula, it is known from the KSA and the UAE (Collingwood, 1985; Collingwood & Agosti, 1996; Collingwood et al., 2011).

Monomorium brunneolucidulumCollingwood & Agosti, 1996

Monomorium brunneolucidulum Collingwood & Agosti, 1996: 343. Oman. Palearctic.

Remarks. In their brief original description of the enigmatic species M. brunneolucidulum from Oman, Collingwood & Agosti (1996) neither gave successful diagnostic characters nor illustrations for species recognition. In addition, the type-material is apparently lost. Due to a lack of type material and species diagnostic characters, it is impossible to confirm the identity of the species. Until the type material of this species is available we prefer to treat it as a nomen dubium.

Monomorium floricola (Jerdon, 1851)

(Figs. 14A–14C)

Figure 14 M. floricola, (A) body in profile; (B) body in dorsal view; (C) head in full-face view, CASENT0922876 (Michele Esposito), www.AntWeb.org, licensed under CC BY 3.0 Unported.

Atta floricola Jerdon, 1851: 107 (w.) India. Indomalaya.

Forel, 1893: 388 (q.m.); Wheeler, 1905: 88 (q.m.); Donisthorpe, 1914: 136 (gynandromorph); Crawley, 1920: 217 (gynandromorph); Wheeler & Wheeler, 1955c: 121 (l.). Combination in Monomorium: Mayr, 1879: 671.

Senior synonym of Monomorium poecilum: Emery, 1894d: 151.

Senior synonym of Monomorium specularis: Mayr, 1879: 671.

Senior synonym of Monomorium cinnabari: Wheeler, 1913: 486.

Senior synonym of Monomorium floreanum: Brown, in Linsley & Usinger, 1966: 175.

Senior synonym of Monomorium angusticlava: Bolton, 1987: 390.

Senior synonym of Monomorium impressum: Bolton, 1987: 390.

Senior synonym of Monomorium floricola furina: Heterick, 2006: 122.

Senior synonym of Monomorium floricola philippinensis: Heterick, 2006: 122.

Material examined. Oman: Dhofar Province: Ayn Dirbat (17.106, 54.453, 207 m, 17.xi.2017, M.R. Sharaf, 4w: KSMA, 1w: CASENT0922876, CASC).

Geographic Distribution. This species was originally described from India. It is a successful tramp species of putative Southeast Asian origin that is widely distributed throughout tropical and subtropical regions worldwide (Deyrup, Davis & Cover, 2000; Wetterer, 2010a). The present material represents a new record for Oman and the Arabian Peninsula.

Monomorium haritheCollingwood & Agosti, 1996

(Figs. 15A–15C)

Figure 15 M. harithe, (A) body in profile; (B) body in dorsal view; (C) head in full-face view, CASENT0913802 (Will Ericson), www.AntWeb.org, licensed under CC BY 3.0 Unported.

Monomorium harithe Collingwood & Agosti, 1996: 347 (w.) Saudi Arabia. Afrotropic. Holotype worker: Saudi Arabia, desert near Najran, 17.533, 44.000, 10.iv.1983, C.A. Collingwood, CASENT0922335, WML, [examined] syn. n.

Monomorium najrane Collingwood & Agosti, 1996: 352 (w.) Saudi Arabia. Afrotropic, Holotype worker: Saudi Arabia, Najran, semi desert, iv.1984, C.A. Collingwood, CASENT0922335, WML, [image examined] syn. n.

Previous records. KSA: Riyadh (24.714, 46.675, 21.i.1980, A.H. Talhouk, 2w); Yemen: Taiz (13.578, 44.018, 20.iii.1993, C.A. Collingwood, 2w).

Remarks. Monomorium harithe was described from the KSA and Yemen, while M. najrane was described from Najran (KSA) near the Saudi-Yemeni borders (Collingwood & Agosti, 1996). The comparison of the type material of the two species reveals a straightforward synonymy. The two species share the following characters: anterior median clypeal margin distinctly concave; scapes distinctly short, when laid back from their insertions failing to reach posterior head margin; mesosoma without standing hairs; metanotal groove feebly impressed but distinct; propodeal dorsum with distinct furrow; mesosoma, petiole and postpetiole finely and densely punctate; petiole and postpetiole each with single pair of back directed hairs; gaster smooth and shining. In addition, the two species share common body measurements (e.g., HW 0.51; SL 0.53).

Geographic Distribution. This Arabian endemic species is only known from the KSA and Yemen (Collingwood & Agosti, 1996).

Monomorium heggyi Sharaf, sp. n.

(Figs. 16A–16C)

Figure 16 M. heggyi sp. n., (A) body in profile; (B) body in dorsal view; (C) head in full-face view, CASENT0746641 (Zach Lieberman), www.AntWeb.org, licensed under CC BY 3.0 Unported.

Holotype pinned worker. KSA: Al Bahah Province: Shada Al A’la Mountain (19.877, 41.311, 897 m, Al Dhafer et al., MRS0261, CASENT0746641, KSMA).

Paratype pinned workers. KSA: Al Baha Province: Shada Al A’la Mountain (19.877, 41.311, 897 m, 04.vi.2014, M. R Sharaf, 4w, 1w: CASENT0746641); Shada Al A’la Mountain (19.863, 41.301, 1,225 m, 08.xii.2014, Al Dhafer et al., 6w, 1w: CASENT0922301); Shada Al A’la Mountain (19.843, 41.312, 1,666 m, 23.viii.2014, Al Dhafer et al., 1w: CASENT0906391); Shada Al A’la Mountain (19.877, 41.311, 897 m, 04.vi.2014, M.R. Sharaf, MRS0261, 2w); Shada Al A’la Mountain (19.877, 41.311, 897 m, 23.viii.2014, M.R. Sharaf, MRS0261, 1w: CASENT0746641); Shada Al A’la Mountain (19.863, 41.301, 1,225 m, 08.xii.2014, Al Dhafer et al., 1w); Shada Al A’la Mountain (19.851, 41.301, 1,325 m, 15.ii.2014, Al Dhafer et al., 1w); Shada Al A’la Mountain (19.839, 41.310, 15.xi.2015, Al Dhafer et al., 3w: KSMA).

Other Material. Jazan Province: Zabia (17.107, 42.650, 43 m, 09.iv.2012, M. R Sharaf, MRS0070, 2w); Abu Arish (17.013, 42.802, 90 m, 10.iv.2012, M. R Sharaf, MRS0073, 10w); Jazan (16.97627, 42.61743, 38 m, 12.iv.2012, M. R Sharaf, MRS0077, 25w: KSMA, 1w: CASC, 1w: BMNH, 1w: WML, 1w: OXUM, 1w: LACM).

Measurements. Holotype: TL 2.01; HL 0.49; HW 0.39; SL 0.42; EL 0.12; EM 0.07; ML 0.59; PW 0.25; PTL 0.11; PTW 0.11; PPL 0.09; PPW 0.09; CI 80; EI 31; SI 108.

Paratype workers: TL 1.50–2.55; HL 0.42–0.49; HW 0.32–0.39; SL 0.28–0.42; EL 0.07–0.12; EM 0.07–0.09; ML 0.45–0.59; PW 0.21–0.31; PTL 0.09–0.14; PTW 0.07–0.11; PPL 0.07–0.09; PPW 0.07–0.09; CI [69–86]; EI 21–31; SI 80–108 (n = 13).

Diagnosis. Monomorium heggyi is diagnosed by the following character combination: scapes when laid back from their insertions just reaching posterior margin of head; head in full-face view with eyes located nearly at midlength; promesonotal outline feebly convex or flat, sloping posteriorly to narrow and shallowly impressed metanotal groove.

Worker. Head. Head in full-face view distinctly longer than broad, with concave posterior margin and shallowly convex sides and feebly concave posterior margin; median portion of clypeus with anterior free margin slightly indented; eyes of moderate size, in profile with convex dorsal sides and straight ventral side, maximum diameter 0.21 x– 0.30 x HW, with 7 ommatidia in longest row; head in full-face view eyes located nearly at midlength of head and just breaking sides; scapes when laid back from their insertions just reaching posterior head margin. Mesosoma. Promesonotal outline feebly convex or flat, slopping posteriorly to narrow and shallowly impressed metanotal groove; propodeal dorsum in profile convex making a continuous curve with propodeal declivity and with defined lateral margins. Petiole. Petiole with high rounded node in profile; subpetiolar process broad and blunt. Postpetiole. Postpetiolar node lower than petiolar node in profile and nearly as broad as petiole in dorsal view. Sculpture. Mandibles feebly longitudinally sculptured; cephalic surface with faint vestiges of superficial reticular patterning, almost entirely effaced, area between frontal carinae finely longitudinally striate; clypeus smooth; entire mesosoma, petiole and postpetiole sharply and densely reticulate-punctate; gastral tergites smooth and shining. Pilosity. Dorsum of head without standing hairs behind the level of the frontal lobes; several pairs of long hairs on the anterior clypeal margin and on mandibles; antennae with dense appressed pubescence; mesosoma, petiole and postpetiole without standing hairs of any description; first gastral tergite without standing hairs except for sparse appressed pubescence; pilosity of remaining gastral tergites restricted to the posterior margins. Color. Uniformly yellow.

Remarks. Monomorium heggyi belongs to the M. salomonis species-group (Bolton, 1987). It is most similar to M. rabirium Bolton, 1987 from Botswana from which it is readily distinguished by the longer scapes (SI 80–108) that reach the posterior head margin in full-face view, the posteriorly shifted eyes located nearly at the midlength of head in full-face view. Monomorium rabirium has shorter scapes (SI 92–97) that fail reaching the posterior head margin in full-face view, and eyes conspicuously located in front of midlength of head in full-face view. Among the Arabian species of the M. salomonis species-group, M. heggyi is superficially similar to M. elghazalyi from the Socotra Archipelago from which it can be easily separated by the larger eyes (EI 21–31), the shallowly impressed metanotal groove, and the densely sculptured mesosoma, petiole and postpetiole. Monomorium elghazalyi has smaller eyes (EI 19–20), broadly and deeply impressed metanotal groove, and a smooth body surface.

Etymology. The patronymic name honors Dr. Essam Heggy, the Egyptian space scientist at NASA.

Habitat. The type locality, Shada Al A’la, is a Nature Reserve (Fig. 17) located in the Al Bahah Province in the southwestern KSA at an elevational range of 470–2,222 m. The locality is characterized by relatively high rainfall, diverse habitats, and high biodiversity, as well as by the presence of large areas of terraced fields used for cultivating banana, coffee, figs, and lemon. The region has a diverse range of wild vegetation cover including plants of the Leguminosae (Fabaceae), composites (Asteraceae), and graminoides (Poaceae). Acacia (Fabaceae) and Juniper (Cupressaceae) are the most dominant plants (SWA, 2018; El-Hawagry et al., 2016). Shada Al-A’Ala harbors a high number of endemic animals including birds, mammals (SWA, 2018) and insects (El-Hawagry et al., 2016).

Figure 17 Shada Al A’la, the type locality of M. heggyi sp. n. (A. Shams Al Ola).

Geographic Distribution. KSA.

Monomorium khalidi Sharaf, sp. n.

(Figs. 18A–18C)

Figure 18 M. khalidi sp. n., (A) body in profile; (B) body in dorsal view; (C) head in full-face view, CASENT0922288 (Michele Esposito), www.AntWeb.org, licensed under CC BY 3.0 Unported.

Holotype pinned worker. KSA: Al Bahah Province: Shada Al A’la (19.839, 41.310, 1,563 m, 18.x.2014, Al Dhafer et al., CASENT0922288, KSMA).

Three paratype, pinned workers, KSA: Jazan Province: Wadi Shahdan (17.452, 42.715, 200 m, 13.xi.2012, M.R. Sharaf, MRS0131, CASENT0919810, KSMA); Fayfa, Wadi Al Jora (17.279, 43.062, 419 m, 06.iv.2013, M.R. Sharaf, KSMA).

Measurements. Holotype: TL 3.16; HL 0.75; HW 0.67; SL 0.62; EL 0.14; EM 0.25; ML 0.95; PW 0.44; PTL 0.27; PTW 0.21; PPL 0.17; PPW 0.21; CI 89; EI 21; SI 93.

Paratype workers: TL 2.31–2.98; HL 0.59–0.72; HW 0.49–0.59; SL 0.45–0.59; EL 0.14–0.15; EM 0.11–0.18; ML 0.71–0.85; PW 0.32–0.41; PTL 0.28–0.32; PTW 0.11–0.15; PPL 0.11–0.17; PPW 0.08–0.15; CI 81–83; EI 24–29; SI 92–102 (n = 3).

Diagnosis. Monomorium khalidi can be distinguished by the combination of the following characters: short scape failing to reach posterior head margin in full-face view; abundant mesosomal pilosity; straight outline of promesonotum; densely reticulate-punctate surfaces of head, mesosoma, petiole, and postpetiole; promesonotum dorsally with at least five to six pair of hairs, promesonotum and propodeum each with three pairs.

Worker. Head. Head nearly as long as broad, or little longer than broad with concave posterior margin and feebly convex sides; median portion of clypeus with anterior free margin distinctly concave; eyes of moderate size, in profile view with convex dorsal sides and straight ventral side, maximum diameter 0.20 × HW, with 9–10 ommatidia in longest row); head in full-face view with eyes failing to break head sides; scapes when laid back from their insertions failing to reach posterior margin. Mesosoma. Promesonotal outline flat, slopping posteriorly to narrow and feebly impressed metanotal groove; propodeal dorsum flat and short, longitudinally concave, with sharply defined lateral margins. Petiole. Petiole with high rounded node in profile. Postpetiole. Postpetiole as broad as petiole in dorsal view. Sculpture. Cephalic surface between and immediately behind frontal lobes finely longitudinally striate; cephalic surface and sides, entire mesosoma, petiole and postpetiole sharply and densely reticulate-punctate; first gastral tergite shagreened and relatively shining. Pilosity. Cephalic surface with several pairs of standing hairs behind level of frontal lobes; posterior margin of head with three pairs of standing hairs; underside of head with about five pairs of short hairs; promesonotum dorsally with at least five to six pair of hairs, promesonotum and propodeum each with three pairs; petiole and postpetiole each with two-three pairs of backward directed hairs; first gastral tergite with numerous standing hairs which are evenly distributed over the sclerite in front of the apical transverse row. Color. Bicolored species, head, mesosoma, petiole, postpetiole and appendage light red brown, gaster black.

Remarks. This new species is a member of the M. salomonis species-group (Bolton, 1987). Monomorium khalidi is closest to M. junodi Forel, 1910 from South Africa in terms of the relatively small eyes, the short scapes that fail to reach posterior margin of head, the acute metanotal groove, and densely punctate body surface. The two species have short scapes that fail to reach posterior margin of head in full-face view, metanotal groove feebly impressed; body surface densely punctate; and propodeal longitudinally concave, with sharply defined lateral margins. However, M. khalidi can be easily distinguished from M. junodi by the following characters: body bicolored with head, mesosoma, petiole, postpetiole and appendages light red-brown contrasting with the black gaster; head in full-face view with eyes failing to break sides; promesonotum dorsally with at least five to six pair of hairs, promesonotum and propodeum each with three pairs; promesonotal outline flat. Monomorium junodi is uniformly brown to dark brown, head in full-face view with eyes breaking sides; promesonotum dorsally with two pair of hairs, propodeum without hairs; promesonotal outline feebly but distinctly convex. Among the Arabian Monomorium species, M. khalidi is superficially similar to M. nitidiventre in terms of body size, surface sculpture, eye shape but the former can be readily recognized by the reduced stiff pilosity.

Etymology. The patronymic name honors Khalid Amr (born at 04/11/2012), the son of the second author.

Habitat. The type locality of M. khalidi is Shada Al A’la (Fig. 19), the same locality where M. heggyi was collected.

Figure 19 Shada Al A’la, the type locality of M. khalidi sp. n. (A. Shams Al Ola).

Geographic Distribution. KSA.

Monomorium niloticumEmery, 1881

(Figs. 20A–20C)

Figure 20 M. niloticum, (A) body in profile; (B) body in dorsal view; (C) head in full-face view, CASENT0905755 (Will Ericson), www.AntWeb.org, licensed under CC BY 3.0 Unported.

Monomorium niloticum Emery, 1881: 533 (w.) Egypt. Palearctic. [MSNG], [Syntype worker, CASENT0905755, image examined].

Combination in Monomorium (Xeromyrmex): Emery, 1922: 179; subspecies of Monomorium venustum: Forel, 1910: 6; Wheeler & Mann, 1916: 170; Stitz, 1917: 346; Finzi, 1936: 175; revived status as species: Santschi, 1936: 37; see also: Collingwood & Agosti, 1996: 352; current subspecies: nominal plus M. n. gracilicorne, M. n. niloticoides.

Senior synonym of Monomorium matame Collingwood & Agosti, 1996: 350, fig. 22 (w.) OMAN. Palearctic, Holotype worker, Oman, Wadi Matam (0l.II.1986, M.D. Gallagher, CASENT0922325, WML) [examined] syn. n.

Material examined. KSA: Asir Province: Jebel Al Habala (18.038, 42.873, W. Buttiker, 1w); Alkhola (13.600, 44.283, 4w, WML). Al Baha Province: Al Atawla, Al Baha-Taif RD, Wadi Bawa (20.750, 41.247, 1,310 m, 08.xi.2012, M.R. Sharaf, MRS0099, 7w, 1w: CASENT0906397); Wadi Elzaraeb (20.073, 41.387, 2,086 m, 09.v.2011, M.R. Sharaf, 3w); Al Mandaq, Wadi Turabah (20.242, 41.263, 1,715 m, 19.ix.2011, M.R. Sharaf, 6w); Hawtat Bani Tamim (23.525, 46.845, 19.iv.2008, M.R. Sharaf, 6w); Riyadh city (06.viii.2008, 1w); Wadi Hanifa (24.671, 46.595, 641 m, 11.iv.2013, M.R. Sharaf, 11w); Al Mandaq, Wadi Turabah (20.211, 41.288, 1,793 m, 10.v.2011, M.R. Sharaf, 4w). Riyadh Province: Al Hayer (24.280, 46.766, 647 m, 10.iii.2011, A.S. Aldawood, 24w); Al Hayer (24.557, 46.744, 589 m, 11.iv.2014, S. Salman, 5w); Dawademi (24.557, 44.377, 983 m, 18.iv.2014, S. Salman, 6w); Dawademi (24.478, 44.364, 1,027 m, 18.iv.2014, S. Salman, 4w); Dawademi (24.583, 44.323, 966 m, 16.i.2015, S. Salman, 9w); Dawademi (24.538, 44.355, 999 m, 16.i.2015, S. Salman, 1w); Afif (23.900, 42.881, 1,052 m, 17.i.2015, 1w); Afif (23.957, 42.976, 1,059 m, 17.i.2015, S. Salman, 6w); Dhurma (24.607, 46.130, 646 m, 30.i.2015, S. Salman, 8w); Hawtat Bani Tamim (23.454, 46.819, 582 m, 19.ii.2015, M.R. Sharaf, 3w); Hawtat Bani Tamim (23.500, 46.850, 612 m, 19.ii.2015, M.R. Sharaf, 2w); Hareeq (23.614, 46.054, 689 m, 22.ii.2015, S. Salman, 29w); Quwayia (24.070, 45.280, 823 m, 03.v.2014, S. Salman, 3w); Majma’a (26.005, 45.019, 730 m, 13.ix.2014, S. Salman, 6w, KSMA); Riyadh (23.953, 43.636, x.1979, W. Buttiker, 2w); Wadi Eflah, x.1983, W. Buttiker, 2w); wadi Mawran (22.050833, 46.671944, 10.ii.1985, W. Buttiker, 6w); Riyadh (24.7136, 46.6753, 07.vii.1975, W. Buttiker, 1w); Shoiba (Shuaibah) (20.6295, 39.5624, 06.xii.1983, W. Buttiker, 2w); Wadi Nimar (24.5705, 46.68, v.1983, W. Buttiker, 2w); Harithi (21.28, 40.28, 11.v.1984, W. Buttiker, 1w); Wadi Ellah (09.ix.1986, W. Buttiker, 3w, WML); Malham (25.154, 46.282, 711 m, 15.ix.2014, S. Salman, 4w); Malham (25.161, 46.229, 742 m, 15.ix.2014, S. Salman, 1w); Quwayia (24.058, 45.245, 846 m, 29.x.2014, S. Salman, 4w); Quwayia (24.05043, 45.25795, 839 m, 29.x.2014, S. Salman, 12w); Quwayia (24.053, 45.262, 836 m, 29.xi.2014, S. Salman, 2w); Na’jan (24.026, 47.138, 467 m, 13.xii.2014, S. Salman, 9w); Wadi Al Dawaser (22.778, 44.786, 686 m, 20.ii.2015, S. Salman, 3w); Wadi Al Dawaser (20.778, 44.786, 686 m, 20.ii.2015, S. Salman, 3w); Qassim, Buraydah (26.216, 44.0414, 633 m, 17.ix.2011, Steyaningrum, 9w); Qassim, Buraydah (26.330, 43.979, 623 m, 19.x.2013, M.R. Sharaf, 9w); Huraymila (25.1487, 45.950, 815 m, 07.ii.2011, M.R. Sharaf, 9w); Dirab Research Station (24.419, 46.654, 570 m, 28.ix.2011, B.L. Fisher, 1w, CASENT0260164; Dirab Research Station (24.737, 46.618, 662 m, 29.ii.2012, K. Mahmoud, 27w); Naam Dam (23.628, 46.631, 646 m, 22.ii.2015, S. Salman, 30w); Dirab (24.409, 46.662, 588 m, 30.xii.2009, M.R. Sharaf, 6w); Salboukh (25.079, 46.347, 689 m, 05.xi.2009, M.R. Sharaf, 15w); Salboukh (25.074, 46.377, 728 m, 26.xi.2014, S. Salman, 12w); Ghiyanah (25.074, 46.226, 728 m, 26.xii.2014, S. Salman, 3w); Al Hayer (24.280, 46.766, 10.iii.2011, A.S. Aldawood, 10w); Qassim, Buraydah (26.338, 44.024, 643 m, 19.x.2013, S. Salman, 2w); Wadi Al Dawaser (20.487, 44.764, 690 m, 22.i.2014, S. Salman, 1w, All previous material in KSMA); Wadi Khumra (17.viii.1979, W. Buttiker, CASENT0249836, 1w, NHMB). Mekkah Province: Ras Hatibah (21.978, 38.937, 11.i.1983, 2w, WML). OMAN: Nakhl (23.44696, 57.88062, 364 m, 02.iv.2016, M.R. Sharaf, 8w, CASENT0922306); Dhofar, Dhalkout (16.727, 53.249, 623 m, 18.xi.2017, M.R. Sharaf, 6w, CASENT0922859, KSMA); no locality (2005, 1w); no locality (xi.1984, 1w); Jebel Akhdar (23.073, 54.662, 1w); Wattayah (23.591, 58.363, 1983, 3w, WML). UNITED ARAB EMIRATES: Wadi Maidaq (25.300, 56.117, 22.xi-02.xii.2010, M. Hauser et al., UAE12977, 1w, CASENT0264568, KSMA); Hatta (24.806, 56.125, iii.1998, A.V. Harten, 2w); Hatta (24.806, 56.125, iii.1995, A.V. Harten, 2w, WML). YEMEN: Ta’iz (13.578, 44.018, A.V. Harten, 1w); Al Kawd (13.089, 45.365, 1991, A.V. Harten, 2w, WML).

Remarks. Monomorium matame was described from Oman and KSA based on the worker caste. In the original description Collingwood & Agosti (1996) pointed out the similarity between M. matame and M. niloticum and used variable characters that were not useful in species recognition. Our examination of the type material of both shows that M. matame is not separable from M. niloticum. The two species share the following characters: scapes when laid back from their insertions failing to reach posterior head margin; eyes relatively large about 0.30–0.33 × HW; metanotal groove deeply impressed; mesosoma with several pairs of scattered standing hairs; promesonotum with three pairs, mesonotum with two to three pairs, propodeum with a single pair of hairs; bicolored species with head, mesosoma, petiole, postpetiole red-brown contrasting with dark brown to black gaster. Based on the examination of the type images of both species, we propose to synonymize M. matame as a junior synonym of M. niloticum, on the basis of morphological similarity.

Geographic Distribution. Monomorium niloticum is originally described from Egypt and widely spread in the Arabian Peninsula (Collingwood, 1985; Collingwood & Agosti, 1996; Collingwood et al., 2011, Sharaf et al., 2018b). It is also collected from Israel (Vonshak & Ionescu-Hirsch, 2009), and North Africa (Sharaf, 2006; Borowiec, 2014). Monomorium niloticum is one of the most broadly spread myrmicine species throughout several countries of the Arabian Peninsula including the KSA, Oman, UAE, and Yemen (Collingwood, 1985; Collingwood & Agosti, 1996).

Ecological and Biological notes. The broad geographic distribution of the species can be interpreted in the light of the species diverse habitat preferences, including the deserts, mountainous, and cultivated sites. Several worker series were found nesting in either dry or humid soil beneath rocks in an undisturbed site in the KSA where a broad diverse of plant species exists including Acacia (Fabaceae), Citrus limon (L.) Osbeck (Rutaceae), Prunus dulcis (Mill.) D. A. Webb (Rosaceae), Juniperus L. (Cupressaceae), Mangifera indica L. (Anacardiaceae), Ficus sp. (Moraceae), Hibiscus L. (Malvaceae) and Azadirachta indica A. Juss. (Meliaceae).

Monomorium nitidiventre Emery, 1893

(Figs. 21A–21C)

Figure 21 M. nitidiventre, (A) body in profile; (B) body in dorsal view; (C) head in full-face view, CASENT0904602 (Will Ericson), www.AntWeb.org, licensed under CC BY 3.0 Unported.

Monomorium bicolor subsp. nitidiventris Emery, 1893: 256 (w.) Egypt. Palearctic. [MSNG], [Syntype worker, CASENT0904602, image examined].

Mayr, 1901: 7 (q.); Karavaiev, 1911: 5 (m.); combination in Monomorium (Xeromyrmex): Wheeler, 1922: 869; subspecies of Monomorium bicolor: Santschi, 1938: 39; of Monomorium subopacum: Santschi, 1927: 245; Menozzi, 1932: 94; Bernard, 1953: 159; Hamann & Klemm, 1967: 413; raised to species: Collingwood, 1985: 272; see also: Wheeler & Mann, 1916: 171; Collingwood & Agosti, 1996: 352.

Monomorium yemene Collingwood & Agosti, 1996: 357, fig. 32 (w.) Yemen. Afrotropic. Holotype worker, Yemen, Taiz (20.x.1991, A. van Harten, NHMB) [Presumed lost]; Paratype worker, Yemen, Zingibar- Shuqrah (13.356, 45.700, 21.iii.1993, C.A. Collingwood, CASENT0913865, NHMB) [examined] syn. n.

Material examined. YEMEN, W. Adem Port, Wadi Tiban, N. W. of Jebel Jihaf (13.198, 44.787, ~1,158 m, 22.x.1937, C.A. Collingwood, from flower of Adenium sp., B. M. Exp. To S. W. Arabia, H. Scott & E. B. Britton, B. M. 1938-246, BMNH (E) 1017382, 1w, CASENT0914158, BMNH).

Remarks. The synonymy of M. yemene with M. nitidiventre is straightforward since both are morphologically similar and indistinguishable. Both share the following key characters: posterior margin emarginated in full-face view; median portion of anterior clypeal margin distinctly concave; metanotal groove deeply impressed; head, mesosoma, petiole, and postpetiole densely reticulate-punctate and covered with abundant pale standing hairs.

Note: However, the locality label of the paratype specimen (CASENT0913865) (Madinat Al shiraq) is not matching the locality data mentioned in the original description but the collecting data and collector are congruous with the description, therefore, this specimen is treated as one of the type material of M. yemene.

Geographic Distribution. Monomorium nitidiventre is originally described from Egypt and recorded from the KSA, Kuwait and Yemen (Collingwood, 1985; Collingwood & Agosti, 1996).

Monomorium subdenticorneCollingwood & Agosti, 1996

(Figs. 22A–22C)

Figure 22 M. subdenticorne, (A) body in profile; (B) body in dorsal view; (C) head in full-face view, CASENT0914318 (Zach Lieberman), www.AntWeb.org, licensed under CC BY 3.0 Unported.

Monomorium subdenticorne Collingwood & Agosti, 1996: 354, fig. 27 (w.) Yemen. Afrotropic.

Material examined. KSA: Asir Province: Ahad Refedah (18.134, 43.001, 2,179 m, 23.ii.2015, M. Alharbi, 2w, KSMA). Yemen: Ghaiman, about 9 miles S. E. of San’a (13.933, 44.833, ~8,400 ft, 18.ii.1938, B. M. Exp. To S. W. Arabia, H. Scott & E. B. Britton, B. M. 1938-246, BMNH1017456, 1w, CASENT0914318, BMNH).

Geographic Distribution. A species originally known from Yemen and herein collected for the first time from the KSA.

Monomorium venustum (Smith, 1858)

(Figs. 23A–23C)

Figure 23 M. venustum, (A) body in profile; (B) body in dorsal view; (C) head in full-face view, CASENT0902221 (Will Ericson), www.AntWeb.org, licensed under CC BY 3.0 Unported.

Myrmica venusta Smith, 1858: 126 (w.) Syria. Palearctic.

Material examined. KSA: Asir Province: Wadi Asidah (20.417, 41.200, 10.ix.1983, 1,480 m, W. Buttiker, 1w, WML). Riyadh Province: Zulfi (26.272, 44.771, 635 m, 18.i.2014, S. Salman, 15w); Mezahmyia (24.466, 46.251, 648 m, 29.xi.2014, S. Salman, 10w); Dawademi (24.552, 43.932, 873 m, 16.i.2015, S. Salman, 1w); Bijadriyah (24.310, 43.731, 439 m, S. Salman, 2w); Afif (23.900, 42.081, 1,052 m, 17.i.2015, S. Salman, 18w); Shaqra (25.270, 45.291, 712 m, 23.i.2015, 63w); Sajir (25.165, 44.601, 750 m, 23.i.2015, S. Salman, 6w); Shaqra (25.274, 45.300, 707 m, 24.i.2015, S. Salman, 2w, KSMA). Kuwait: Kuwait (no date, 2w, WML). Oman: Wahiba sands (21.438, 58.554, 23.iii.1986, W. Buttiker, 1w, WML). United Arab Emirates: Jebel Hafit (24.050, 55.767, 27.ii-03.iii.2011, M. Hauser et al., UAE13010, 1w, CASENT0264463); Um Al-Quwain (12. iv-07.vi.2009, M. Hauser et al., UAE12920, 1w, CASENT0264584); Ar-Rafah (25.717, 55.867, 15-30. x.2010, M. Hauser et al., UAE12866, CASENT0264475, KSMA); Wadi Asidah (20.417, 41.200, 10.ix.1983, W. Buttiker, 1w); Riyadh (24.714, 46.675, 18.ii.1975, W. Buttiker, 1w); Sharjah (25.346, 55.421, vii.2003, A.V. Harten, 2w, WML). Yemen: Sana’a (15.369, 44.191, iii.1990, 1w, WML).

Geographic Distribution. Monomorium venustum is originally described from Syria and recorded from the KSA, Kuwait, Oman (Collingwood, 1985; Collingwood & Agosti, 1996; Sharaf et al., 2018b), Israel (Vonshak & Ionescu-Hirsch, 2009), and North Africa (Borowiec, 2014).

Biogeographical analysis

Twenty-four species are Arabian endemics. Afrotropical (5) and Palearctic (12) species were also identified (Table 1; Figs. 24–29).

Table 1 Biogeography of the Arabian Monomorium. References: 1 = Collingwood (1985), 2 = Bolton (1987), 3 = Collingwood & Agosti (1996).

Species	Type locality	Bioregion	Reference	Distribution map*	
M. abeillei	ISRAEL	Palearctic	1; 2; 3	Fig. 24	
M. acutinode	OMAN	Palearctic	3	–	
M. aeyade	OMAN	Endemic	3; Sharaf et al. (2018a)	–	
M.areniphilum	TUNISIA	Palearctic	1; 3	–	
M. asiriense	KSA	Endemic	3	–	
M. barbatulum	KAZAKHSTAN	Palearctic	1; 3	Fig. 25	
M. bicolor	ERITREA	Afrotropic	1; 2; 3	Fig. 25	
M. brunneolucidulum	OMAN	unknown	3	–	
M. buettikeri	KUWAIT	Endemic	3	–	
M. buxtoni	IRAQ	Palearctic	1; 3	–	
M. carbo	ETHIOPIA	Afrotropic	1; 3	–	
M. carbonarium	MADEIRA	Palearctic	3	–	
M. clavicorne	ISRAEL	Palearctic	3; Sharaf et al. (2018a)	–	
M. dammame	KSA	Endemic	3	–	
M. dirie	OMAN	Endemic	3	–	
M. elghazalyi	YEMEN	Endemic	Sharaf et al. (2017a)	–	
M. exiguum	ETHIOPIA	Afrotropic	3; Sharaf et al. (2018a)	–	
M. fayfaense	KSA	Endemic	3	–	
M. floricola	INDIA	Tramp	Heterick (2006)	Fig. 25	
M. gallagheri	OMAN	Endemic	3	–	
M. hanaqe	KSA	Endemic	3	–	
M. harithe	KSA	Endemic	3	Fig. 26	
M. heggyi	KSA	Endemic		Fig. 26	
M. holothir	KENYA	Afrotropic	2	–	
M. jizane	KSA	Endemic	3	–	
M. khalidi	KSA	Endemic		Fig. 26	
M. knappi	YEMEN	Endemic	3	–	
M. luteum	YEMEN	Afrotropic	1; 3	–	
M. mahyoubi	YEMEN	Endemic	3	–	
M. moathi	YEMEN	Endemic	Aldawood, Sharaf & Collingwood (2010)	–	
M. mohammedi	KSA	Endemic	3; Sharaf et al. (2018a)	–	
M. niloticum	EGYPT	Palearctic	1; 3	Fig. 27	
M. nimihil	YEMEN	Endemic	Collingwood et al. (2004)	–	
M. nitidiventre	EGYPT	Palearctic	1; 3	Fig. 26	
M. pharaonis	EGYPT	Tramp	1; 3	–	
M. rimae	YEMEN	Endemic	3	–	
M. riyadhe	KSA	Endemic	3	–	
M. salomonis	EGYPT	Palearctic	1; 3	–	
M. sarawatense	KSA	Endemic	3; Sharaf et al. (2018a)	–	
M. subdenticorne	YEMEN	Endemic	3	Fig. 28	
M. subopacum	PORTUGAL	Palearctic	1; 2; 3	–	
M. suleyile	KSA	Endemic	3	–	
M. tumaire	KSA	Endemic	3	–	
M. venustum	SYRIA	Palearctic	1; 3	Fig. 29	
Note:

* Maps were created for species with coordinate records.

Figure 24 Distribution map of M. abeillei.

Figure 25 Distribution map of M. barbatulum, M. bicolor, M. floricola.

Figure 26 Distribution map of M. harithe, M. khalidi sp. n., M. nitidiventre, M. heggyi sp. n.

Figure 27 Distribution map of M. niloticum.

Figure 28 Distribution map of M. subdenticorne.

Figure 29 Distribution map of M. venustum.

Discussion

Monomorium is one of the most diverse ant genera in the world, but it is rarely the most speciose genus on a regional scale. However, the Arabian Monomorium fauna, based on review of previous literature data and on our current work, includes 44 species, making it the most diverse known ant genus of the Arabian Peninsula (Collingwood, 1985; Collingwood & Agosti, 1996; Collingwood et al., 2011; Aldawood & Sharaf, 2011; Sharaf & Aldawood, 2013b; Sharaf et al., 2015, 2017a, 2018a, 2018b). Currently, the genus represents ~14% of the total number of species reported from the region (312 spp.) (Collingwood, 1985; Collingwood & Agosti, 1996; Collingwood et al., 2011; Aldawood & Sharaf, 2011; Sharaf & Aldawood, 2013c, 2019; Sharaf et al., 2015, 2017a, 2017b, 2018a, 2018b; Sharaf, Aldawood & Hita Garcia, 2019; Sharaf et al., 2020c). This value is lower than the 20% mentioned by Collingwood & Agosti (1996). This reduction is the result of numerous taxa in previous studies now treated as junior synonyms of other species (Sharaf et al., 2015, 2017a, 2018a; this study).

We found that ~55% of the Monomorium species (24/44) appear to be endemic to the Arabian Peninsula. High degrees of endemism have been reported for many of Arabian arthropod groups, including ants in general (Collingwood, 1985; Collingwood & Agosti, 1996; Sharaf, Al Dhafer & Aldawood, 2014; Sharaf, Fisher & Aldawood, 2014; Sharaf et al., 2017a, 2018b), staphylinid beetles (Assing et al., 2013; Hlaváč, Sharaf & Aldawood, 2013), carabid beetles (Abdel-Dayem et al., 2018), termites (Cowie, 1989), lepidopterans (Larsen, 1984), and pseudoscorpions (Mahnert, Sharaf & Aldawood, 2014).

Biogeographically, the biota of the Arabian Peninsula does not constitute a cohesive unit (Larsen, 1984; Sharaf et al., 2020a, 2020b). Instead, the Arabian Peninsula is often considered to be at the nexus of two terrestrial biogeographic realms, the Palearctic and the Afrotropic. In fact, Olson et al. (2001) places the northern and central Arabian Peninsula in the Palearctic bioregion (along with Europe, northern Africa, Asia north of the Himalayas, and neighboring islands) and the southern and eastern coasts of the Arabian Peninsula in the Afrotropic bioregion (along with sub-Saharan Africa, southern Iran, southwestern Pakistan, and neighboring islands). Our Geographic distribution data (see also Table 1) of Arabian Monomorium supports this basic division, with 12 Palearctic species more common in the north and central deserts and the five Afrotropic species more common in the southern region and along the coasts.

The majority of the endemic Arabian ant species has faunal similarities with taxa from the Afrotropic bioregion that has been earlier documented by several studies (Larsen, 1984; Collingwood, 1985; Collingwood & Agosti, 1996; Sharaf & Aldawood, 2011, 2019; Sharaf, Aldawood & Taylor, 2012; Sharaf, Aldawood & El-Hawagry, 2012a, 2012b, Sharaf et al., 2020a, 2020b; El-Hawagry et al., 2013, 2017; Hájek & Reiter, 2014). Therefore, it is not surprising that a large proportion of the endemic Arabian Monomorium species (13/24) have been found in the mountainous ranges of southwestern KSA that extend to Yemen.

Both new Monomorium species reported here, M. heggyi and M. khalidi, were collected in the Shada Al-A’Ala Nature Reserve (SANR), a protected area consisting of an isolated granite mountain massif in southwestern Saudi Arabia. Its location, elevational range (470–2,222 m) and high rainfall resulting in diverse microclimates and a high biodiversity (SWA, 2020). As a unique biodiversity hotspot, the SANR contains ~495 plant species (~22% of the total reported Saudi Arabian flora), including 43% of the threatened plant species and 19 endemic plants (Thomas, El-Sheikh & Alatar, 2017). The SANR also protects a diverse fauna, including rare and endemic vertebrates, including the griffon vulture (Gyps fulvus (Hablitz)), the Arabian leopard (Panthera pardus nimr (Hemprich and Ehrenberg)), and the Arabian wolf (Canis lupus arabs Pocock) (SWA, 2020). The SANR invertebrate fauna has attracted relatively little attention, but recent insect biodiversity inventories and monitoring research projects conducted by King Saud University Museum of Arthropods resulted in two important faunistic studies that recorded 119 Diptera species (El-Hawagry et al., 2016) and 62 carabid beetle species (Abdel-Dayem et al., 2019). Further studies are planned to be carried out at SANR to explore additional levels of biodiversity.

In addition to the native Monomorium species, there are two Monomorium known from the Arabian Peninsula that are cosmopolitan tramp species, spread around the world through human commerce: M. pharaonis and M. floricola. The pharaoh ant, M. pharaonis, is a common domestic pest. Although it was first described from Egypt, its original native range is uncertain (Wetterer, 2010b). We report the first known Arabian record of M. floricola, an Indomalayan species, from a single site in Oman. Although widespread around the world, M. floricola is rarely considered a serious pest. However, because this species is very small, slow moving, cryptically colored, and primarily arboreal, its abundance and ecological importance may be underappreciated (Wetterer, 2010a). Human activities, population movements, and global trade around the world greatly contribute to the further spread of tramp species outside their native habitats (Sharaf et al., 2020c, 2020d).

Considering the high degree of endemism encountered, it is likely that the known Arabian Monomorium fauna will increase in the future with further exploration of poorly surveyed areas of the Arabian Peninsula, especially the southwestern mountains of the KSA, Yemen, and the mountainous regions of Oman and the UAE. We hope that the present study will serve as a cornerstone of future taxonomic treatments of Monomorium in the Arabian Peninsula.

The authors thank Barry Bolton for useful discussions about Monomorium over years. We are indebted to the following colleagues: Boris Kondratieff for useful comments, Brian Fisher for kind permission to use Monomorium images on AntWeb, Michele Esposito for imaging the new species, Mahmoud Abdel-Dayem for creating the distribution maps, and Ahmed Shams Al ‘Ola for technical assistance with figures. Special thanks to Brian Heterick and Kadri Kiran for valuable suggestions.

Additional Information and Declarations

Competing Interests

Author Contributions

Data Availability

New Species Registration

The authors declare that they have no competing interests.

Mostafa R. Sharaf conceived and designed the experiments, performed the experiments, analyzed the data, prepared figures and/or tables, authored or reviewed drafts of the paper, and approved the final draft.

Amr A. Mohamed conceived and designed the experiments, performed the experiments, analyzed the data, prepared figures and/or tables, authored or reviewed drafts of the paper, and approved the final draft.

Brendon E. Boudinot performed the experiments, analyzed the data, authored or reviewed drafts of the paper, and approved the final draft.

James K. Wetterer performed the experiments, analyzed the data, prepared figures and/or tables, authored or reviewed drafts of the paper, and approved the final draft.

Francisco Hita Garcia conceived and designed the experiments, performed the experiments, analyzed the data, prepared figures and/or tables, authored or reviewed drafts of the paper, and approved the final draft.

Hathal M. Al Dhafer performed the experiments, authored or reviewed drafts of the paper, and approved the final draft.

Abdulrahman S. Aldawood conceived and designed the experiments, performed the experiments, analyzed the data, authored or reviewed drafts of the paper, and approved the final draft.

The following information was supplied regarding data availability:

All data supporting the work results are presented in the in the figures and Materials and Methods and Results sections.

The specimens/material described in the manuscript are stored at King Saud University Museum of Arthropods. Specimen numbers are: CASENT0922263, CASENT0922876, CASENT0746641, CASENT0922288, CASENT0919810, CASENT0922859, CASENT0264568, CASENT0264475.

The following information was supplied regarding the registration of a newly described species:

Publication LSID:

urn:lsid:zoobank.org:pub:A7FFDF5C-6CD5-41CA-B106-BFF6BDFCB258

Monomorium heggyi sp. n. LSID:

urn:lsid:zoobank.org:act:B57EC2EA-1781-4C19-ADFE-757C834E2774

Monomorium khalidi sp. n. LSID:

urn:lsid:zoobank.org:act:3B5BB529-D842-4146-B8F7-ACCAC9CD5BA7.

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
