# Peer review of "Monomorium (Hymenoptera: Formicidae) of the Arabian Peninsula with description of two new species, M. heggyi sp. n. and M. khalidi sp. n"

_PeerJ, doi:10.7717/peerj.10726_

## Round 0.1 · original submission · Minor Revisions

Dear Dr. Sharaf and colleagues:

Thanks for submitting your manuscript to PeerJ. I have now received two independent reviews of your work, and as you will see, the reviewers raised some relatively minor concerns about the research. This is great and indicates optimism for your work and the potential impact it will have on research studying ants of the Arabian Peninsula.

While the concerns of the reviewers are relatively minor, this is a major revision to ensure that the original reviewers have a chance to evaluate your responses to their concerns. There are many suggestions, which I am sure will greatly improve your manuscript once addressed.

Please note that both reviewers kindly provided marked-up versions of your manuscript.

Therefore, I am recommending that you revise your manuscript, accordingly, taking into account all of the issues raised by the reviewers. I do believe that your manuscript will be greatly improved once these issues are addressed.

Good luck with your revision,

-joe

·

Basic reporting

This taxonomic paper follows a number of similar papers on the ants of the Arabian Peninsula produced by the senior author, often, as here, with the assistance of co-contributors. The paper has a pleasingly professional feel about it, and follows a good, logical sequence. I particularly liked the photographs of ant specimens that illustrate key diagnostic features. I feel this is a very helpful aid to the reader. The English language format is smooth, without major irregularities or awkward phraseology, but, as is usual in a paper with many technical details, I noticed a number of small errors that I have highlighted on the .pdf. I have also included pop-up notes here and there with suggestions for the authors. The bibliography is comprehensive, but I was unable to find some of the listed authors cited in the actual text. In terms of the article structure, I am not convinced that a separate section on ‘Biogeographical analyses (V.) is needed, as it seems to me that the information contained under it could easily be discussed for each species under ‘Geographic Distribution’.

Experimental design

This is a standard, well-integrated taxonomic paper. The taxonomy is of the older, morphological variety but the taxonomic keys and diagnoses appear sound, and in keeping with what I know of the genus covered (Monomorium). The authors have correctly followed the new species policies, and their work meets the ICZN standard and the published work and the nomenclatural acts have been registered in Zoobank. The key to the workers of Arabian Monomorium is easy to follow; in just a couple of cases it could be tweaked to cover exceptions in one or other of the couplet lugs (most notably the first lug of couplets 1 and 2). In terms of more substantive issues, one taxon, namely Monomorium barbatulum, looks to me like a Trichomyrmex, and has the broad head and metasomal configuration of the latter genus. Heightening my suspicions, I note that a specimen whose image appears in AntWeb was identified as a Trichomyrmex by the senior author in 2016. I suggest the authors revisit their interpretation of this ant. I have also queried whether Monomorium wahibiense should be synonymised under Monomorium abeillei, but I leave that to the authors’ discretion.

Validity of the findings

All data have been provided. The species’ descriptions could be checked for standardisation of the way synonymies are described in the text, and I noticed ‘Syn. n.’ for ‘syn. n.’ on a couple of occasions. The captions for the figures are truncated, and I am wondering if this has been brought about by the process of electronic transmission. I also query the duplication of captions for the diagnostic images of the workers in three orientations (as is done in AntWeb). I note that this duplication is not identical, as ‘sp. n.’ is bolded in the second entry but not the first.
All in all, I consider this paper an important contribution to our taxonomic understanding of the ants from this region.

Additional comments

No comment.

·

Basic reporting

No comment.

Experimental design

No comment.

Validity of the findings

No comment

Additional comments

Dear Author,
Your manuscript has very important information about the taxonomy of the Arabian Monomorium genus. Pictures are of very high quality. Descriptions are suitable and large enough. New species and other taxonomic acts are suitable.
I suggested some small changes to the text.

---

## Round 0.2 · Minor Revisions

Dear Dr. Sharaf and colleagues:

Thanks for revising your manuscript. The reviewer is mostly satisfied with your revision (as am I). Great! However, there are a few issues still to entertain. Please address these ASAP so we may move towards acceptance of your work.

Please note that while the reviewer stated “A small handful of typographical errors still present are highlighted in the revised MS, and an annotated copy is attached with this review” we have been unable to receive this file from the reviewer. Thus, move forward with your revision without it, and please check your writing for any typos.

Best,

-joe

·

Basic reporting

No change from previous report.

Experimental design

No change from previous report.

Validity of the findings

No change from previous report.

Additional comments

I am pleased to see most of my recommendations have been followed. However, I would still like to see a justification for Monomorium barbatulum being included within the genus. (I accept this may be warranted but, based on my previous comment, its close resemblance to a Trichomyrmex and an earlier placement of a specimen under that genus should be noted.) A small handful of typographical errors still present are highlighted in the revised MS, and an annotated copy is attached with this review.

---

## Round 0.3 · accepted · Accept

Dear Dr. Sharaf and colleagues:

Thanks for revising your manuscript based on the concerns raised by the reviewers. I now believe that your manuscript is suitable for publication. Congratulations! I look forward to seeing this work in print, and I anticipate it being an important resource for groups studying ants of the Arabian Peninsula. Thanks again for choosing PeerJ to publish such important work.

Best,

-joe